

# Diagnostic efficiency of multi-modal MRI based deep learning with Sobel operator in differentiating benign and malignant breast mass lesions—a retrospective study

Weixia Tang[1,*], Ming Zhang[1,*], Changyan Xu[2], Yeqin Shao[2], Jiahuan Tang[1], Shenchu Gong[1], Hao Dong[3] and Meihong Sheng[1]

[1] Radiology Department, Affiliated Hospital 2 of Nantong University, Nantong First People's Hospital, NanTong, Jiangsu, China
[2] School of Transportation and Civil Engineering, Nantong University, Nantong, China
[3] Department of Research Collaboration, R&D Center, Beijing Deepwise & League of PHD Technology Co., Ltd., Beijing, China
[*] These authors contributed equally to this work.

Corresponding authors
Shenchu Gong,
gongshenchu@msn.com
Meihong Sheng, smh4127@163.com

## ABSTRACT

**Purpose**. To compare the diagnostic efficiencies of deep learning single-modal and multi-modal for the classification of benign and malignant breast mass lesions.
**Methods**. We retrospectively collected data from 203 patients (207 lesions, 101 benign and 106 malignant) with breast tumors who underwent breast magnetic resonance imaging (MRI) before surgery or biopsy between January 2014 and October 2020. Mass segmentation was performed based on the three dimensions-region of interest (3D-ROI) minimum bounding cube at the edge of the lesion. We established single-modal models based on a convolutional neural network (CNN) including T2WI and non-fs T1WI, the dynamic contrast-enhanced (DCE-MRI) first phase was pre-contrast T1WI (d1), and Phases 2, 4, and 6 were post-contrast T1WI (d2, d4, d6); and Multi-modal fusion models with a Sobel operator (four_mods:T2WI, non-fs-T1WI, d1, d2). Training set ($n = 145$), validation set ($n = 22$), and test set ($n = 40$). Five-fold cross validation was performed. Accuracy, sensitivity, specificity, negative predictive value, positive predictive value, and area under the ROC curve (AUC) were used as evaluation indicators. Delong's test compared the diagnostic performance of the multi-modal and single-modal models.
**Results**. All models showed good performance, and the AUC values were all greater than 0.750. Among the single-modal models, T2WI, non-fs-T1WI, d1, and d2 had specificities of 77.1%, 77.2%, 80.2%, and 78.2%, respectively. d2 had the highest accuracy of 78.5% and showed the best diagnostic performance with an AUC of 0.827. The multi-modal model with the Sobel operator performed better than single-modal models, with an AUC of 0.887, sensitivity of 79.8%, specificity of 86.1%, and positive prediction value of 85.6%. Delong's test showed that the diagnostic performance of the multi-modal fusion models was higher than that of the six single-modal models (T2WI, non-fs-T1WI, d1, d2, d4, d6); the difference was statistically significant ($p = 0.043, 0.017, 0.006, 0.017, 0.020, 0.004$, all were greater than 0.05).

**Conclusions**. Multi-modal fusion deep learning models with a Sobel operator had excellent diagnostic value in the classification of breast masses, and further increase the efficiency of diagnosis.

## INTRODUCTION

Breast cancer has become the most common malignant tumor in women (*Ghoncheh, Pournamdar & Salehiniya, 2016*) and the second leading cause of cancer-related deaths (*DeSantis et al., 2017*). Early detection, diagnosis, and treatment are important for the treatment and prognosis of breast cancer (*Pharoah et al., 2013*; *Lauby-Secretan et al., 2015*; *Myers et al., 2012*). Breast MRI has been widely used in clinical work for diagnosis, disease staging and postoperative evaluation of neoadjuvant chemotherapy (*Zhang et al., 2019*; *Monticciolo, 2017*).

Breast cancer is a highly heterogeneous tumor (*Santamaría et al., 2010*; *Baltzer et al., 2010*), and information about this heterogeneity is difficult for traditional radiologists to capture with the naked eye. An artificial intelligence (AI)-assisted system, based on breast MRI, extracts valuable and subtle information from the lesion image, such as, lesion size, shape, signal intensity, and texture, covering all the features of the degree of tumor heterogeneity.

To date, there are three primary methods for AI-assisted disease diagnosis, which are the texture analysis, radiomics, and deep learning (*McNitt-Gray et al., 1999*; *Lv et al., 2017*; *Xu et al., 2022*). Several studies have been conducted on texture analysis and radiomics. These methods, based on the two dimensional (2D) or 3D texture and radiomics characteristics of the lesion, have good performance in differentiating the benign and malignant lesions, classifying the molecular states, and predicting the treatment efficacy (*Pharoah et al., 2013*; *Lauby-Secretan et al., 2015*; *Myers et al., 2012*). However, both these methods require high-throughput extraction of image features, feature screening, dimensionality reduction, and modeling analysis, to complete the tasks, such as, classification, regression, and clustering. It is necessary to simultaneously verify their diagnostic performance. The entire process is time-consuming and yields many characteristic parameters, which are difficult to analyze (*Meyer-Base et al., 2021*; *Tariciotti et al., 2022*; *Gillies, Kinahan & Hricak, 2016*).

Deep learning belongs to the category of machine learning. It uses multilevel nonlinear information processing and abstraction to perform supervised or unsupervised feature learning, representation, classification, and pattern recognition. Deep learning can provide accurate results for image segmentation, classification, and feature extraction (*Chan, Samala & Hadjiiski, 2020*). The most commonly used deep learning method is the convolutional neural network (CNN), which extracts image features and automatically builds the models. Studies have shown that deep learning has significantly more potential in various clinical tasks, such as, image segmentation, lesion detection, classification, and

diagnosis (*Anwar et al., 2018*). *Truhn et al. (2019)* compared the effectiveness of CNN and radiomics in the classification of benign and malignant breast enhancement lesions based on the 2D features of the enhanced subtraction images. The results showed that the performance of CNN was better than that of radiomics, and as the sample size increased, the performance of CNN increased. Most previous studies on deep learning used a single sequence, and there have been few studies on the diagnostic efficacy of different sequences or multi-sequence fusion. The multi-modal feature guidance module is used to reduce the difference between different modal features so that the modal information can be better integrated (*Wang, Cui & Zhu, 2020*; *Zhuo et al., 2022*). *Zhuo et al. (2022)* proposed a deep learning-based multi-modal feature fusion network. Through three-branch networks, US images, SWE, and CDUS images of the thyroid are combined to diagnose thyroid nodules. Experiments show that the proposed method can effectively complete the diagnosis of thyroid nodules through images of three modalities, and its results are better than that of a single modality and two modalities. In our reserarch, we established single-modal models, including T2WI and non-fs T1WI. The dynamic contrast-enhanced (DCE-MRI) first phase was non-enhancement T1WI-fs, and Phases 2, 4, and 6 was enhanced as well as the multi-modal fusion model (T2WI, non-fs-T1WI, Pre-contrast T1WI, first post-contrast T1WI series fusion). The diagnostic efficacies of the models in the classification of benign and malignant breast masses were evaluated to provide new insights into the differential diagnosis of breast mass lesions.

# MATERIALS AND METHODS

## Dataset

We collected data from patients who underwent breast magnetic resonance imaging (MRI) examinations at the Nantong First People's Hospital from January 2014 to October 2020 retrospectively. This study was approved by Nantong First People's Hospital Ethics Committee (Ethics number: 2022KT146). All the volunteers signed an informed consent form before the inspection. **Inclusion criteria**: ①The lesion diameter was more than one cm or at least two consecutive slices were visible to the naked eye; ②The image quality was good without distinct artifacts or distortion; ③All the lesions were mass lesions. **Exclusion criteria**: ①Breast mass showing no enhancement on DCE-MRI; ②Patient had a history of surgery or radiotherapy/chemotherapy before the breast MRI; ③Pathological diagnosis of the lesion was unclear. In total, 296 patients with breast mass lesions were enrolled. According to the inclusion and exclusion criteria, the following patients were excluded: seven patients with unqualified image quality, 12 patients with a history of puncture or mastectomy before the breast MRI examination, 45 patients with small breast lesions and multiple small enhancement nodules in the breast, 14 patients with incomplete examination sequence or breast MRI perfusion scan, and 15 patients with breast mass lesions and non-mass enhancement lesions. Finally, 203 patients with 207 tumors were enrolled in the study. The average patient age was ($48.5 \pm 13.1$) years in the range 17–86 years. Among them, there was only one male patient aged 54 years. There were 105 patients with malignant tumors, with an average age of ($55.5 \pm 11.3$) years in the range

**Table 1  Age and pathological details of patients with breast lesions.**

|  | Benign case ($n = 98$) | Malignant case ($n = 105$) |
|---|---|---|
| Age (years) | 41.0 ± 10.6 | 55.5 ± 11.3 |
| Pathological result | Fibrocystic adenosis ($n = 18$) | Invasive ductal carcinoma ($n = 81$) |
| (207 masses) | Fibroadenoma ($n = 71$) | Breast mucinous adenocarcinoma ($n = 7$) |
|  | Intraductal papilloma ($n = 4$) | Invasive lobular carcinoma ($n = 3$) |
|  | Intraductal papilloma ($n = 4$) | Intraductal carcinoma or tubular carcinoma ($n = 8$) |
|  | Benign lobular carcinoma ($n = 4$) | Solid papillary carcinoma ($n = 2$) |
|  |  | Junctional phylloides tumor ($n = 1$) |
|  |  | Small lymphocytic lymphoma ($n = 1$) |
|  |  | Medullary carcinoma ($n = 3$) |

**Table 2  Breast MRI scan parameters.**

| Abbreviations | MR sequence | Scan parameters |
|---|---|---|
| T2WI | T2 axis reversal recovery sequence | TR 4300 ms, TE 61 ms, Reversal angle 80°, Layer thickness/spacing 4 mm/4.4 mm |
| Non-fs T1WI | T1 Axial non-fat suppression three-dimensional spoiler gradient echo sequence | TR 6.04 ms, TE 2.45 ms, Reversal angle 20°, Layer thickness/spacing 1.4 mm/0 mm |
| DWI | Axial plane echo imaging sequence | TR 7300 ms, TE 82 ms, Reversal angle 90°, $b = 50, 400, 800$ s/mm², Layer thickness/spacing 4 mm/4.4 mm |
| DCE-MRI Phase 16 | Dynamic enhanced T1 axial fat suppression three-dimensional spoiler gradient echo sequence | TR 4.67 ms, TE 1.66 ms, Reversal angle 10°, FOV 340 mm × 340 mm, Layer thickness/spacing 1.2 mm/0 mm, Phase 1 was non-enhancement scan, Phase 2–6 were enhanced scan |

32–86 years; 98 patients had benign tumors, aged 17–65 years, with an average age of (41.0 ± 10.6) years; among all the mass lesions, 106 were malignant and 101 were benign (Table 1). Data were collected as previously described in *Sheng et al. (2021)*.

## Breast MRI: equipment and method

MRI scans were performed using a Siemens 3.0T MRI scanner (Verio: Siemens, Erlangen, Germany) with a 16-channel phased-array coil specified for the breast. The patient was placed in the prone position with the head entering the scanner first. The patient's breasts naturally hung in the breast coil, and the nipple remained at the center of the coil. The scan sequence parameters were as follows (Table 2): DCE T1-weighted axial fat suppression 3D spoiler gradient echo: TR 4.67 ms, TE 1.66 ms, flip angle 10°, FOV 340 mm × 340 mm; slice thickness 1.2 mm, scanning of six phases without interval; scan time 6 min 25 s; high-pressure syringe injection of 15–20 mL contrast agent Gd-DTPA based on body weight (0.2 ml/kg) at a flow rate of two mL/s, followed by an injection of the same amount of normal saline to flush the tube. After the 25 s injection, scanning was triggered and each phase was collected for 1 min. The DCE-MRI Phase 1 was non-enhanced named pre-contrast series, and Phases 2–6 were enhanced. Phase 2 images were named DCE-MRI T1WI first post-contrast sequence.

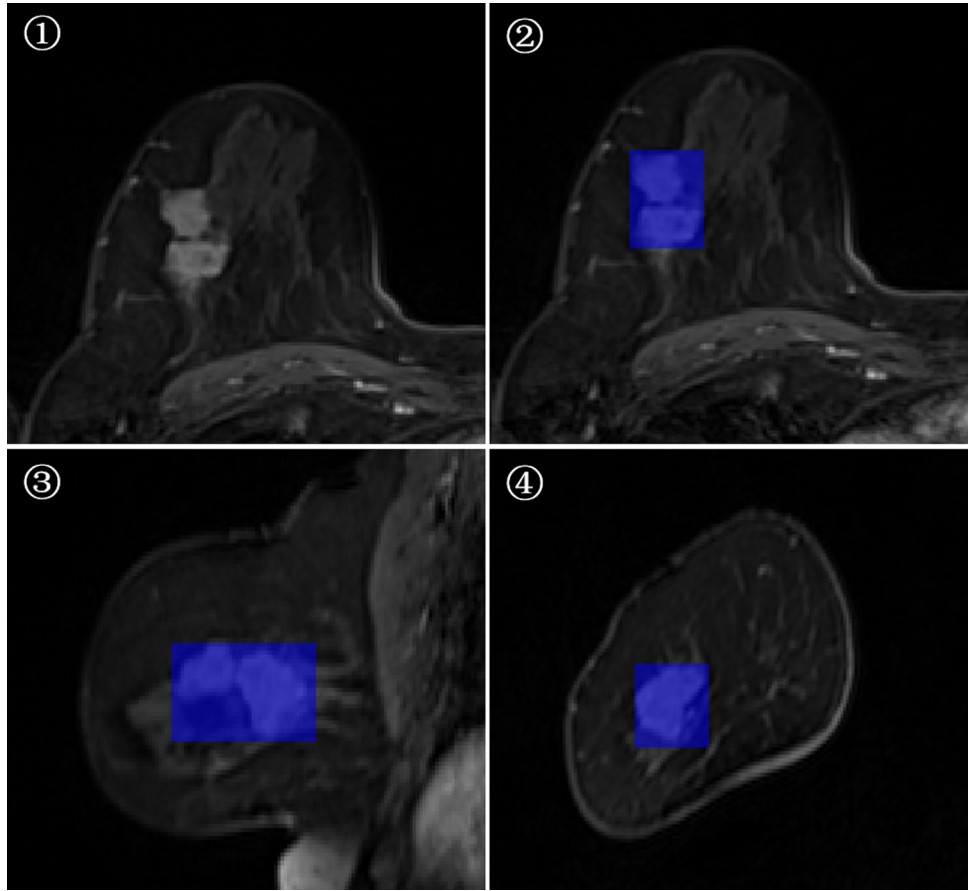

**Figure 1** **Minimum bounding cube segmentation method based on the post-contrast T1WI series.**

## Image segmentation

Images of T2WI, non-fs-T1WI, and DCE-MRI T1WI Phases 1, 2, 4, and 6 in the breast MRI examination were selected, and the DICOM format images were imported into the processing software (ITK-SNAP 3.8.0). All the lesions were manually segmented by a resident doctor with eight years of experience in breast MRI diagnosis and reviewed by an associate chief physician with more than 10 years of experience in breast MRI diagnosis. Disagreements were resolved through consultation.

3D-ROI segmentation based on the lesion edge was used to select the largest lesion section on three coordinates (axial, coronal, and sagittal), and the smallest box, which could cover the outermost edge of the tumor was plotted (Figs. 1 and 2). Cystic degeneration, necrosis, and calcification within the lesions were unavoidable. Using a computer traversal algorithm, the 3D coordinates of the voxel points with non-zero intensity values in the segmented image were obtained, and the minimum and maximum values in each dimension were calculated. Using the matrix slice package in Python, a cube containing the lesion could be "cut out" from the original image.

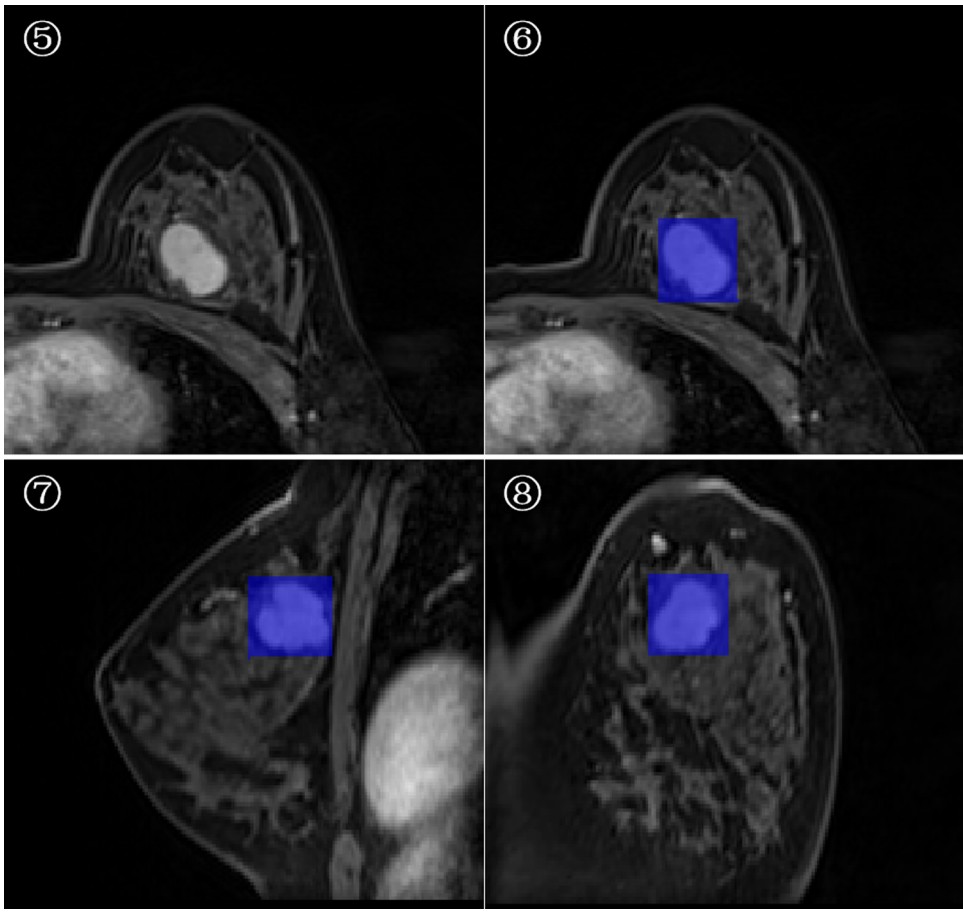

**Figure 2** **Minimum bounding cube segmentation method based on the post-contrast T1WI series.**

## Image processing and deep learning method

The data were pre-processed before the training process to increase the convergence speed of the algorithm. Before loading the MRI images into the network, the intensity value of the input data was 0-1 normalized. The normalization formula is as follows:

$$x = \frac{X - X_{\min}}{X_{\max} - X_{\min}}$$

where $x$ represents the normalized image pixel value, $X$ represents the original image pixel value, and $X_{\max}$ and $X_{\min}$ represent the maximum and minimum pixel values of the smallest circumscribed cube, respectively.

In this study, 207 mass lesions were analyzed. We obtained three datasets: the training set ($n = 145$), validation set ($n = 22$), and test set ($n = 40$). The model was trained on the training set, the validation set was used to verify the model during training, and the test set was used to evaluate the performance of the model after the training was completed.

The deep learning model was implemented using the TensorFlow framework and Python. Similar to ResNet (*Khalili & Wong, 2018*; *Sheng et al., 2021*), the main modules of the network included the input modality, 3D-resmodule, 3D-CBAM (convolutional

block attention module), down-sampling module, and fully connected module. The residual module was used to extract the features. It consisted of a 3D convolution layer, a batchnorm layer, an activation function, and the ReLU layer. A short connection was added at the end to speed up the network convergence and prevent gradient explosion. 3D-CBAM was used to find the key features at the channel level and spatial level and to assign higher weights to them. 3D-CBAM mainly used average pooling, maximum pooling and softmax functions to calculate these weights. The downsampling module is key to building a deep network as it can reduce the size of the feature map and increase the number of feature map channels, thereby reducing the calculation cost while retaining the main information. The downsampling module was mainly composed of 3D convolutional layers, which were used to adjust the convolution kernel size, stride size, and number of output channels. In Fig. 2, the square volume is used to reflect the size change in the feature map. After the last 3D-CBAM, the algorithm stretched the high-dimensional output features into a 1D vector, connected two fully connected layers, and finally obtained two values (*P1, P2*), which represented the probabilities of benign and malignant lesions. *P1* > *P2* indicated that the lesion was benign, and *P1* ≤ *P2* indicated that the lesion was malignant.

To obtain the maximum possible information from the different modalities and reduce the speed of network convergence, the strategy of a multi-modal fusion network is to increase the channel number of the network input tensor, that is, to splice the aforementioned four modalities in the channel dimension. The input tensor of the multi-modal network included T2WI, non-fs-T1WI, pre-contrast T1WI, and the first post-contrast T1WI series. Because the non-fs T1WI sequence could appropriately show the edge features of most mass lesions, such as, edge smoothing, burrs, and lobes, we have included non-fs-T1WI scans.

To further increase the input information of the multi-modal network, Sobel gradient images for each modality were added. The Sobel operator can effectively extract the information of the edge point on the image with sharp intensity changes and suppress the non-edge points, thereby improving the edge information of the lesion and reducing the information redundancy caused by using the minimum bounding cube.

For 2D images, the Sobel operator used a 3 × 3 filter to filter the image in the vertical and horizontal directions to obtain the corresponding gradient image. The two sets of filter matrices and gradient calculation methods are as follows.

$$G_x = \begin{bmatrix} -1 & 0 & 1 \\ -2 & 0 & 2 \\ -1 & 0 & 1 \end{bmatrix} * A \text{ and } G_y = \begin{bmatrix} 1 & 2 & 1 \\ 0 & 0 & 0 \\ -1 & -2 & -1 \end{bmatrix}$$

where $G_x$ represents the results of the horizontal calculation, $G_y$ represents the results of the vertical calculation, and A represents the image matrix. $G_x$ and $G_y$ can be used to approximate the gradient value of each pixel using the following formula:

$$G = \sqrt{G_x^2 + G_y^2}.$$

The formula for calculating the gradient direction is as follows:

$$\theta = \tan^{-1}\left(\frac{G_y}{G_x}\right).$$

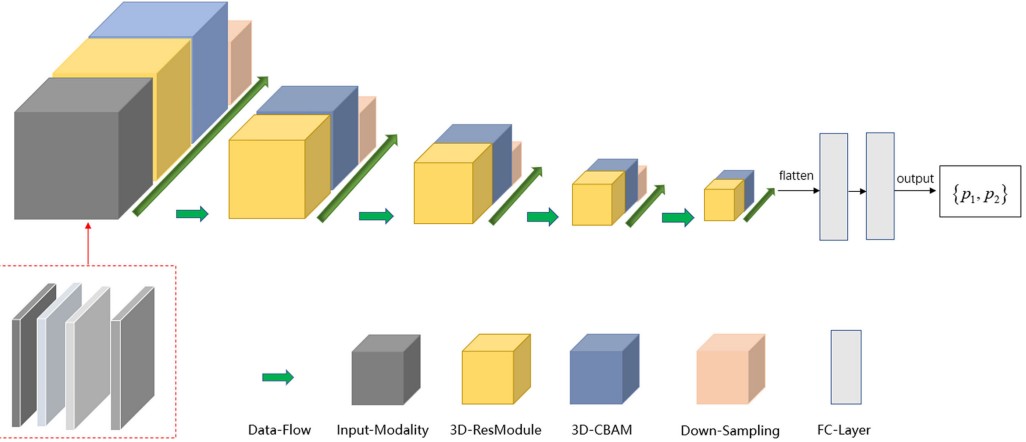

**Figure 3** Schema of multi-modal network channel with Sobel operator.

**Table 3 Tumor classification network parameters.**

| Layer_name | Input_size | Detailed_parameters | Output_size |
|---|---|---|---|
| conv | 8*64*64*64 | kernel = 3, stride = 1, padding = 1 | 8*64*64*64 |
| res_conv*2 | 8*64*64*64 | kernel = 3, stride = 1, padding = 1 | 8*64*64*64 |
| 3D_CBAM | 8*64*64*64 | 1*1*1 convolution | 8*64*64*64 |
| conv | 8*64*64*64 | kernel = 2, stride = 2, padding = 0 | 16*32*32*32 |
| res_conv*4 | 16*32*32*32 | kernel = 3, stride = 1, padding = 1 | 16*32*32*32 |
| 3D_CBAM | 16*32*32*32 | 1*1*1 convolution | 16*32*32*32 |
| conv | 16*32*32*32 | kernel = 2, stride = 2, padding = 0 | 32*16*16*16 |
| res_conv*4 | 32*16*16*16 | kernel = 3, stride = 1, padding = 1 | 32*16*16*16 |
| 3D_CBAM | 32*16*16*16 | 1*1*1 convolution | 32*16*16*16 |
| conv | 32*16*16*16 | kernel = 2, stride = 2, padding = 0 | 64*8*8*8 |
| res_conv*4 | 64*8*8*8 | kernel = 3, stride = 1, padding = 1 | 64*8*8*8 |
| 3D_CBAM | 64*8*8*8 | 1*1*1 convolution | 64*8*8*8 |
| conv | 64*8*8*8 | kernel = 2, stride = 2, padding = 0 | 128*4*4*4 |
| res_conv*4 | 128*4*4*4 | kernel = 3, stride = 1, padding = 1 | 128*4*4*4 |
| 3D_CBAM | 128*4*4*4 | 1*1*1 convolution | 128*4*4*4 |
| Flatten | 128*4*4*4 | | 8192 |
| Full connection | 8192 | | 2048 |
| Full connection | 2048 | | 512 |
| Softmax | 512 | | 2 |

In the next step, we performed edge detection on the images to improve the deep network classification performance. We used the Sobel edge detection image filter in the simple ITK library to process the images. Because each modality had its own Sobel gradient image, the number of input channels of the network was increased from four to eight (Fig. 3). Five-fold cross validation was performed; the detailed parameters are listed in Table 3.

## Model evaluation and statistical analysis

The diagnostic performance of single-modal and multi-modal deep learning models in classifying benign and malignant breast tumors was evaluated using the receiver operating characteristic curve (ROC), area under the curve (AUC), accuracy, sensitivity, specificity, positive predictive value, and negative predictive value. The criteria for using AUC to evaluate the model discrimination ability were as follows (*Alba et al., 2017*): ①AUC < 0.6: poor discrimination; ②0.6 < AUC < 0.75: a certain degree of discrimination; ③AUC > 0.75: good discrimination. Delong's test compared the diagnostic performances of the multi-modal and single-modal models. The model test results are displayed in the form of histograms.

## RESULTS

In this study, we established six single-modal models, which included T2WI, non-fs T1WI, DCE-MRI pre-contrast T1WI (d1), and post-contrast T1WI (d2, d4, d6), and multi-modal fusion models with Sobel operator (four_mods: T2WI, non-fs-T1WI, d1, d2) (Table 4). The deep learning models based on the breast MRI had high degrees of discrimination in the classification tasks, and the AUC values were all greater than 0.75. Among the single-modal models, T2WI, non-fs-T1WI, d1, and d2 had specificities of 77.1%, 77.2%, 80.2%, 78.2%, respectively. d2 had the highest accuracy of 78.5% and showed the best diagnostic performance with an AUC of 0.827. The multi-modal model with the Sobel operator performed better, with an AUC of approximately 0.887, sensitivity of 79.8%, specificity of 86.1%, and positive prediction value of 85.6%. ROC curves and classification histograms of the seven different models are shown in Fig. 4. The score values of the modeling results in the original data table are displayed in intervals. Each sample obtains a corresponding score value after the model calculation, and the score values of all the samples are counted. The histogram shows that the classification score values of the samples were mainly concentrated on both the sides, indicating that this result was excellent. The Delong test showed that the diagnostic performance of multi-modality (fusion) was higher than that of the six single modalities (T2WI, non-fs-T1WI, d1, d2, d4, d6); the difference was statistically significant ($p = 0.043, 0.017, 0.006, 0.017, 0.020, 0.004$, all values were greater than 0.05).

## DISCUSSION

The 3D-ROI segmentation method, based on the minimum bounding cube at the lesion edge, is relatively simple. Deep learning CNN can learn a large number of features. Some features are not important for the final result and some play a key role in predicting the results. Based on the radiologist's reading habits, the models assigned higher weights to the key features according to the advantages of each MR sequence. Therefore, we established an attention mechanism that integrated both time and space.in this study . For example, the models assigned more hemodynamic characteristics to the DCE-MRI model, emphasized tumor heterogeneity information in T2WI, and concentrated on the lesion edge information in the non-fs T1WI.

**Table 4  Performance of six single-modal models and the multi-modal fusion models with Sobel operator.**

| Evaluation index | T2WI | Non-fs-T1WI | d1 | d2 | d4 | d6 | Four_mods |
|---|---|---|---|---|---|---|---|
| AUC | 0.806 | 0.793 | 0.785 | 0.827 | 0.792 | 0.778 | 0.887 |
| Sensitivity (%) | 75.0 | 75.9 | 71.1 | 78.8 | 75.0 | 79.8 | 79.8 |
| Specificity (%) | 77.1 | 77.2 | 80.2 | 78.2 | 78.2 | 70.3 | 86.1 |
| NPV (%) | 75.0 | 75.7 | 72.9 | 78.2 | 75.2 | 77.2 | 80.6 |
| PPV (%) | 77.2 | 77.5 | 78.7 | 78.8 | 78.0 | 73.5 | 85.6 |
| Accuracy (%) | 76.1 | 76.6 | 75.6 | 78.5 | 76.6 | 75.1 | 82.9 |
| Precision (%) | 77.2 | 77.4 | 78.7 | 78.8 | 78.0 | 73.4 | 85.6 |
| Recall (%) | 75.0 | 76.0 | 71.2 | 78.8 | 75.0 | 79.8 | 79.8 |

**Notes.**
d1, d2, d4, and d6 refer to DCE-MRI Phase 1 (DCE-MRI Pre-contrast T1WI), Phase 2 (the first post-contrast T1WI series), Phase 4, and Phase 6 (delayed enhancement after injection); Four_mods refer to the multi-modal fusion model (T2WI, non-fs-T1WI, Pre-contrast T1WI, the first post-contrast T1WI series fusion) with the Sobel operator.

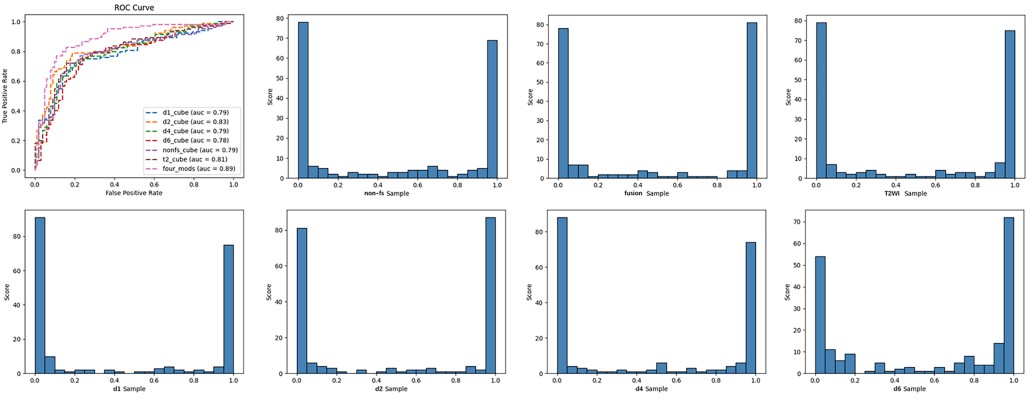

**Figure 4  ROC curves and classification histogram for seven different models.**

## Efficiencies of single-modal deep learning models based on different sequences in the classification of benign and malignant breast tumors

The AUC of the T2WI fat suppression sequence was 0.806, only second to the post-contrast T1WI; sensitivity was 75.0%, and specificity was relatively low at approximately 77.1%. The reason might be that the T2WI images reflect tumor heterogeneity, that is, they can exhibit free water components in tissues and organs. For example, the histological feature of fibroadenoma is that the extracellular matrix contains a large amount of hyaluronic acid and some mucoglycan acid, and an increase in the water content elevates the T2WI signal (*Kuhl et al., 1999*). Malignant tumors usually lack fat content, which is more distinct in some cases where fat degeneration leads to a diffuse increase in normal tissue signals, such as, lobular carcinoma, ductal carcinoma, fibrosis of tissues around the lesion, dense cells, and/or high nucleoplasm ratio; all these lead to reduced T2WI signal (*Kuhl et al., 1999*).

The single-modal model based on non-fs T1WI had a specificity of 77.2% and an AUC of 0.793. Studies have shown that this sequence can sufficiently display the features of the

lesion edge, such as, smooth or ragged edges, and the surrounding fat infiltration. These features are important to determine the malignancy of a lesion. Therefore, this sequence emphasizes the weight of the lesion edge information and significantly improves the model performance.

The deep learning models based on the four DCE-MRI single modalities showed good discrimination ability in classification tasks. This result was related to the hemodynamic information of the lesion reflected by the DCE. DCE-MRI has high values in diagnosing breast cancer and evaluating the disease conditions and treatment efficacy (*Li et al., 2019*). The morphology of the time-intensity curve (TIC) based on DCE is usually divided into three types: inflow type (Type I), platform type (Type II), and outflow type (Type III). *Hetta (2015)* classified the benign and malignant lesions based on the curve morphology with a sensitivity of 80% and specificity of 73.3%. *Dalmis et al. (2016)* calculated the dynamic characteristics of DCE (such as, time of maximum slope, time to enhancement, and final slope) and found an AUC of 0.841. By studying the shape, edge, enhancement characteristics, and involvement of the surrounding tissue in the DCE image, *Wang, Guang & Zhang (2017)* identified the distinct differences between the benign and malignant breast lesions. However, the diagnostic specificity of traditional DCE-MRI remains relatively low (*Wang, Guang & Zhang, 2017*). In this study, we found that the differential diagnosis specificity of the single-modal model based on DCE-MRI first post-contrast T1WI was 78.2% and the AUC was 0.827. The diagnostic efficiency of this model was significantly improved, which might be related to the degree of early disease enhancement (*Sheng et al., 2016*). Early enhancement is closely related to the abundance of blood vessels and blood perfusion during the disease. Malignant lesions usually grow rapidly; they have thick blood vessels and multiple blood vessels, which are often immature. Therefore, malignant lesions have a large number of arteriovenous anastomoses and their early enhancement is more distinct.

## Efficiency of multi-modal fusion deep learning model in the classification of benign and malignant breast tumors

In this study, we used a multi-modal fusion model to study the classification efficiency by combining conventional MRI and DCE-MRI. *He et al. (2015)* found that the edge of the lesion, the type of time signal curve, and the apparent diffusion coefficient (ADC) had greater differential diagnostic significance, and the combination improved the accuracy of MR diagnosis of breast masses.

However, we excluded diffusion-weighted imaging (DWI) and ADC in this study. The main reasons are as follows. First, the quality of DWI and ADC images is affected by equipments with low resolution, which may lead to decreased accuracy when segmenting the lesion. Second, *Dietzel et al. (2021)* established a Kaiser scoring model based on T2WI and DCE-MRI to compare with ADC, and the results showed that both could distinguish the benign and malignant lesions efficiently. However, the diagnostic power of ADC was slightly lower than that of the Kaiser score model, with AUC of 0.811 and 0.909, respectively. Moreover, the Kaiser scoring model was twice as likely to avoid unnecessary

biopsy of benign breast lesions as ADC, and combining ADC with the Kaiser scoring model did not improve the diagnostic performance.

Therefore, we included T2WI, non-fs-T1WI, DCE-MRI pre-contrast, and the first post-contrast T1WI single modalities to establish a multi-modal deep learning model. According to the reading habits of radiologists, information on the edge of the tumor, such as, smooth, lobulated, and burr, plays a significant role in the differentiation of benign and malignant lesions. Therefore, we used the Sobel operator to effectively extract the information of edge points on the images with sharp intensity changes and suppress the non-edge points, thereby improving the edge information of the lesion and reducing the information redundancy caused by using the minimum bounding cube. The AUC of the multi-modal model with the Sobel operator was approximately 0.887, with sensitivity of 79.8%, and specificity of 86.1%. This study demonstrated that the multi-modal fusion model with Sobel could significantly improve the ability of the model to classify the benign and malignant lesions and provide good clinical guidance for inexperienced physicians.

In this study, the multi-modal fusion model diagnosed one case of benign phyllodes tumor and one case of intraductal papillomatosis as malignant. The reason for this analysis may be that this case is a phyllodes tumor, and edema signals can be seen in the peritumoral area, which is enhanced. Later, it became heterogeneous and distinctly enhanced, and it was difficult to distinguish it from malignant tumors. Two cases of invasive ductal carcinoma were misdiagnosed as benign and one case of fibroadenoma was diagnosed as malignant. The common feature of these lesions is their relatively small size, which may lead to the limited extraction of lesion features. In the follow-up, we need to perform large sample size and multi-center verification to improve the model.

## LIMITATIONS

This was a retrospective study with a small sample size and lack of external verification. In future studies, we will consider model verification using an independent test set to assess the actual predictive ability of the model. This study did not include clinical indicators to establish the model because the purpose of this study was to explore the performance of a multi-modal MR deep learning model in the classification of breast mass lesions, and we will further evaluate the clinical features in the follow-up studies. This study only explored breast mass lesions, and non-mass enhancement lesions were not discussed. We will attempt to include this aspect in future studies.

## CONCLUSION

In summary, the multi-modal deep learning model based on the minimum bounding cube of the tumor edge had a good diagnostic value for the classification of breast mass lesions. The Sobel operator could effectively extract the lesion edge information and further increased the model performance in the classification task.

## ACKNOWLEDGEMENTS

We would like to thank the colleagues of the First People's Imaging Department of Nantong, the teachers of Nantong University and the Deepwise technology team for their academic support in carrying out this research.

### Funding

This study was supported by the Scientific Research Project of the Nantong Municipal Health Committee (QA2021008) and the Jiangsu Province Maternal and Child Health Research Project (F202037). The funders had no role in study design, data collection and analysis, decision to publish, or preparation of the manuscript.

### Grant Disclosures

The following grant information was disclosed by the authors:
Scientific Research Project of the Nantong Municipal Health Committee: QA2021008.
Jiangsu Province Maternal and Child Health Research Project: F202037.

### Competing Interests

The authors declare there are no competing interests. Hao Dong is employed by Beijing Deepwise & League of PHD Technology Co., Ltd.

### Author Contributions

- Weixia Tang conceived and designed the experiments, performed the experiments, analyzed the data, prepared figures and/or tables, authored or reviewed drafts of the article, and approved the final draft.
- Ming Zhang conceived and designed the experiments, performed the experiments, analyzed the data, prepared figures and/or tables, authored or reviewed drafts of the article, and approved the final draft.
- Changyan Xu performed the experiments, analyzed the data, authored or reviewed drafts of the article, and approved the final draft.
- Yeqin Shao analyzed the data, prepared figures and/or tables, and approved the final draft.
- Jiahuan Tang analyzed the data, prepared figures and/or tables, and approved the final draft.
- Shenchu Gong analyzed the data, authored or reviewed drafts of the article, and approved the final draft.
- Hao Dong analyzed the data, performed the computation work, prepared figures and/or tables, and approved the final draft.
- Meihong Sheng analyzed the data, performed the computation work, authored or reviewed drafts of the article, and approved the final draft.

## Ethics

The following information was supplied relating to ethical approvals (*i.e.,* approving body and any reference numbers):

This study was approved by the Ethics Committee of Nantong First People's Hospital, and informed consent was obtained from all the patients.

## Data Availability

The original measurement results are available in the Supplementary File.

The code is available at GitHub and Figshare: https://github.com/BCClfer/Classification-of-benign-and-malignant-lesions.

Weixia, Tang (2023). code.rar. figshare. Journal contribution. https://doi.org/10.6084/m9.figshare.23500557.v1.

## Supplemental Information

Supplemental information for this article can be found online at http://dx.doi.org/10.7717/peerj-cs.1460#supplemental-information.

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
