# Peer review of "Diagnostic efficiency of multi-modal MRI based deep learning with Sobel operator in differentiating benign and malignant breast mass lesions—a retrospective study"

_PeerJ Computer Science, doi:10.7717/peerj-cs.1460_

## Round 0.1 · original submission · Major Revisions

Revise according to the comments of reviewers.

Reviewer 1 ·

Basic reporting

The main work of this paper is to compare the diagnostic efficiency of single-mode and multi-mode classification of benign and malignant breast mass lesions. The topic selection of this paper is relatively appropriate and the viewpoint is correct, but it lacks original ideas, and the argumentation content is sufficient, but the argumentation depth is lacking. Meanwhile, it is deficient in the following aspects. I hope the author can modify it to improve the overall readability.

Experimental design

1. The data set selected by the author has too few pictures. It is hoped that transfer learning or data enhancement operations can be attempted to enhance the number of data sets, so as to better evaluate the model.
2. It is suggested to highlight the good results in the table to improve the visualization of the paper and help readers to better understand it.

Validity of the findings

3. Suggest the author to view the journal template, if you want to compare the results of the table and figure are placed in the results section, so that it can be easy for readers to understand.
4. As mentioned in the discussion section, unimodal and multimodal forms establish an attentional mechanism, which can be elaborated on a little more, for example, by describing its functions.
5. It is hoped that comparative experiments can be added to prove why Sobel gradient is added

Reviewer 2 ·

Basic reporting

The authors compare the diagnostic efficiencies of deep learning single-modal and multi-modal for the classification of benign and malignant breast mass lesions. The experimental results show that multi-modal fusion deep learning models with Sobel operator had excellent diagnostic value in the classification of breast masses, with an AUC of 0.887, sensitivity of 79.8%, specificity of 86.1 %, and positive prediction value of 85.6 %. The paper is well-structured and logical. However, I think the innovation point of this paper is not enough, meanwhile, the credibility of the experiment needs to be further confirmed. Next, let me enumerate these problems:

1. For abbreviations, the full name should be given when it first appears, and abbreviations should be used thereafter. For example, MRI is abbreviated in the Introduction Section, but its full name appears later in the Materials and Methods Section.

2. In Section 1.4, the 'where' leading sentence after the formula is in the same paragraph as the formula, so it should not be preceded by a space.

3. The formula alignment in the manuscript is not uniform. In Section 1.4, the first formula is aligned to the left, while the other formulas are displayed in the middle, and it is recommended to unify the alignment method.

4. The variable should use italics. For example, in Section 1.4, the variables ' X ' and ' x ' do not use italics.

5. Part of the image quality is poor, the font is too small, and difficult to identify. For example, Figure 4.

Experimental design

1. There are grammatical and expressive flaws in the manuscript. For example, the word "Figures" is used in the caption of Figure 1, but as a single figure, the word "Figure" should be used, as is the case with Figure 2.

2. Please perform more analysis on Figures 1 and 2.

3. Please give a reasonable explanation as to why you chose a five-fold cross-validation instead of a ten-fold cross-validation.

Validity of the findings

I suggest the authors to make a comprehensive investigation on the multi-modal methods in the literature in the introduction part and give the analysis to the existing works such as (https://doi.org/10.1016/j.compbiomed.2022.106164,https://doi.org/10.1016/j.compbiomed.2022.106457,https://doi.org/10.1016/j.compbiomed.2020.103823,https://doi.org/10.1016/j.compbiomed.2022.105877,https://doi.org/10.1016/j.compbiomed.2022.106088) to make the whole work more in-depth.

Additional comments

no comment

---

## Round 0.2 · Minor Revisions

Please make revisions to the paper based on the reviewer's comments.

Reviewer 1 has requested that you cite specific references. You may add them if you believe they are especially relevant. However, I do not expect you to include these citations, and if you do not include them, this will not influence my decision.

Reviewer 1 ·

Basic reporting

The author has solved all my problems last time, but there is still a lack of relevant introduction in the introduction.
In the introduction, it is suggested that the authors conduct a comprehensive survey of intelligent algorithms and neural network methods in the literature and analyze existing work, such as(doi.org/10.1016/j.compbiomed.2022.106269, DOI: 10.1093/bib/bbac524. https://doi.org/10.1016/j.isci.2023.106231,)

Experimental design

Reasonable experimental design.

Validity of the findings

This paper results are valid.

Reviewer 2 ·

Basic reporting

From the response letter, the paper has been well revised, and the current version of the manuscript is acceptable for publication.

Experimental design

no comment

Validity of the findings

no comment

Additional comments

no comment

---

## Round 0.3 · accepted · Accept

Following the reviewers' comments and your response, and after full consideration, it is decided to accept.

Reviewer 1 ·

Basic reporting

no comment

Experimental design

no comment

Validity of the findings

no comment

Additional comments

The author has solved all the problems and I agree to the publication of this paper.

Reviewer 2 ·

Basic reporting

From the response letter, the paper has been well revised, and the current version of the manuscript is acceptable for publication.

Experimental design

no comment

Validity of the findings

no comment

Additional comments

no comment